



# Assessing land elevation in the Ayeyarwady Delta (Myanmar) and its relevance for studying sea level rise and delta flooding

Katharina Seeger[1], Philip S. J. Minderhoud[2,3,4], Andreas Peffeköver[1], Anissa Vogel[1], Helmut Brückner[1], Frauke Kraas[1], Nay Win Oo[5], and Dominik Brill[1]

[1]Institute of Geography, University of Cologne, Albertus-Magnus-Platz, 50923 Cologne, Germany
[2]Soil Geography and Landscape group, Wageningen University, Droevendaalsesteeg 3, 6708 PB Wageningen, The Netherlands
[3]Department of Civil, Environmental and Architectural Engineering, University of Padova, Via Marzolo 9, 35131 Padova, Italy
[4]Department of Subsurface and Groundwater Systems, Deltares Research Institute, Daltonlaan 600, 3584 BK Utrecht, The Netherlands
[5]East Yangon University, East Yangon University Road, Thanlyin Township, Thanlyin 11291, Myanmar

*Correspondence to*: Katharina Seeger (k.seeger@uni-koeln.de)

**Abstract.** With their low lying, flat topography, river deltas and coastal plains are extremely prone to relative sea level rise and other water related hazards. This calls for accurate elevation data for flood risk assessments, especially in the densely populated Southeast Asian deltas. However, in data-poor countries such as Myanmar, where high accuracy elevation data is not accessible, often only global satellite based digital elevation models (DEMs), suffering from low vertical accuracy and remote sensing artefacts, can be used by the public and scientific community. As the lack of accurate elevation data hampers the assessment of flood risk, studying available information on land elevation and its reliability is essential, particularly in the context of sea level rise impact. Here, we assess the performance of ten global DEMs in the Ayeyarwady Delta (Myanmar) against the new, local, so called AD-DEM, which was generated based on topographical map elevation data. To enable comparison, all DEMs were converted to a common vertical datum tied to local sea level. While both CoastalDEM v2.1 and FABDEM, perform comparably well, showing the highest correspondence in comparison with AD-DEM and low elevation spot heights, the FABDEM outperforms the CoastalDEM v2.1 by the absence of remote sensing artefacts. The AD-DEM provides a high accuracy, open source and freely available, independent elevation dataset suitable for evaluating land elevation data in the Ayeyarwady Delta and studying topography and flood risk at large scale, while small scale investigations may benefit from a FABDEM locally improved with data from the AD-DEM.

Based on latest IPCC projections of sea level rise, the consequences of DEM selection for assessing the impact of sea level rise in the Ayeyarwady Delta are shown. We highlight the need for addressing particularly low lying populated areas within the most seaward districts with risk mitigation and adaptation strategies while also more inland delta population should be made aware to face a higher risk of flooding due to relative sea level rise in the next ~100 years.





**1) Generation of a local DEM**

*Local spot heights* → *Interpolation* → *Local DEM*

*2015 and 2020 flood maps* — *River inundation*

**2) Accuracy assessment of global DEMs**

*Global DEM*

**Area below future sea level**

**3) Assessment of area and population exposed to SLR**

SRTM   ACE2   ASTGTM v003   AW3D30   TanDEM-X 30m
Copernicus DEM   FABDEM   CoastalDEM v2.1   GLL-DTM   AD-DEM

*Population density* — *LandScan Global 2020 population data*

# 1 Introduction

Deltas are hotspots of human settlement and economic production by providing various ecosystem services. Human-environment interactions are particularly dynamic in Southeast Asian deltas where human activities in form of resource extraction and land use changes are increasing and lead to substantial landscape modifications (e.g., Woodroffe et al., 2006; Myat Myat Thi et al., 2012; Nicholls et al., 2020). At the same time, these low lying and densely populated coastal areas are among the most vulnerable in facing the risk of sea level rise (SLR), extreme waves such as storm surges, and other climate

related extremes (Nicholls and Cazenave, 2010; Williams, 2013; Masson-Delmotte et al., 2021). Against this background, increasing population and land use and land cover change not only challenge the sustainability of deltas to ensure income and





food security for societies in coastal areas (e.g., Hanh Tran et al., 2015) and maintain their functioning in balance with natural dynamics (Ottinger et al., 2013; Renaud et al., 2013; Loucks, 2019), but also contribute to increasing vulnerability (Syvitski, 2008; Tessler et al., 2015; Vogel et al., 2022). For example, the extraction of groundwater, sediment, and other resources as

well as the construction of dams and reservoirs have already reduced sediment supply in major Southeast Asian rivers, such as the Mekong or the Red River (e.g., Wang et al., 2011; Van Binh et al., 2020), and accelerated sediment compaction within the deltas significantly (Syvitski, 2008 and references therein; Minderhoud et al., 2017; Minderhoud et al., 2018; Zoccarato et al., 2018). Consequences are enhanced coastal erosion (due to sediment starvation) and accelerated deltaic subsidence (due to accelerated compaction as well as buildings and infrastructural loading), thereby often exacerbating relative sea level change

and highlighting the critical role of land elevation for low lying coastal areas prone to flooding from SLR and other water related hazards.

Given their multiple informative benefits, digital elevation models (DEMs) have been considered as fundamentals in concepts of vulnerability and risk indices (e.g., McLaughlin and Cooper, 2010; Furlan et al., 2021) and constitute primary physical input data for vulnerability and risk mappings/models (e.g., Fereshtehpour and Karamouz, 2018; Rincón et al., 2018; Nirwansyah

and Braun; 2021). Therefore, when studying natural hazards in a holistic manner, the selection of the DEM is key and significantly shapes the outcome of their assessment (van de Sande et al., 2012; Chen et al., 2021; Xu et al., 2021).

In coastal areas, where the usage of DEMs becomes highly relevant in the context of flood risk assessment, several studies investigated land elevation in the recent past by (i), accuracy assessments of commonly available DEMs (e.g., Du et al., 2016; Hawker et al., 2019; Zhang et al., 2019), and (ii) the generation of global coastal lowland DEMs from globally available data

(Kulp and Strauss, 2018; Vernimmen et al., 2020) or DEMs based on local data (e.g., El-Quilish et al., 2018; Minderhoud et al., 2019). Accurate elevation data are needed to generate realistic assessments of flood exposure, and consequently risk and vulnerability (Gesch et al., 2018). This holds particularly true for coastal lowlands and river deltas, where changes in land elevation exacerbate the impact of relative SLR (RSLR) and where subsidence is dominant (e.g., Becker et al., 2020; Nicholls et al., 2021; Shirzaei et al., 2021). Especially for deltas, characterised by high population and socio-economic growth (Hanson

et al., 2011; Bucx et al., 2014; Szabo et al., 2016; Loucks, 2019), the increasing flood vulnerability requires a comprehensive flood risk assessment, including the impact of RSLR. In this context, the choice of a DEM, being representative for the physical characteristics of a study area, is not trivial and different DEMs may lead to significantly different results and implications in view of flood risk and potentially affected coastal population (Kulp and Strauss, 2019). However, although called for in scientific research for several years now (e.g., Gesch, 2009; Schumann and Bates, 2018; Kulp and Strauss, 2019), high quality

(and ideally ground truth) data are often not available to the public, and thus the establishment of local DEMs and corresponding quality assessments often remain challenging in areas with sparse or hardly accessible data.

In countries such as Myanmar, often only global satellite based DEMs are available to the scientific and public community (cf. GERICS and KfW Development Bank, 2015; Chen et al., 2020; Mulhern, 2020). However, when studying RSLR, the use of such global DEMs is problematic for three reasons: Firstly, their considerably large vertical errors of several metres affect

the quality of the SLR impact assessment, particularly in low lying areas where changes occur on decimetre scale (cf. Gesch,





2013; Schumann et al., 2014; 2018; Minderhoud et al., 2019). Secondly, the elevation data of these DEMs is given either in height above the global geoid or ellipsoid. However, local sea level often differs from the global geoid reference and geoid models differ by up to several metres offset from each other. Consequently, the neglection of datum conversion can lead to additional errors and a misjudgement of the flood risk. Thirdly, remote sensing artefacts, such as stripes, speckle noise, voids,

and other errors (Hirt et al., 2018), not only affect DEM quality itself but also result in erroneous impact assessments (e.g., Wechsler, 2007; Yamazaki et al., 2017; Hirt et al., 2018).

In Myanmar, most flood risk assessments either only consider elevation data indirectly (Kittiphong Phongsaphan et al., 2019), do not specify the DEM used (Khin Thandar Win et al., 2014) or rely on global DEMs such as SRTM, HydroSHEDS, and AW3D to investigate the impact of flooding related to enhanced precipitation (ICHARM et al., 2016; Hein Min Htet, 2017)

and storms (Liu et al., 2017). The Coastal Risk Screening tool of Climate Central, Inc. (https://coastal.climatecentral.org/, last access: 31 August 2022) constitutes a convenient, user-friendly interactive map that visualises the impact of future SLR and extreme water levels based on the recently developed and updated CoastalDEM v2.1 (Kulp and Strauss, 2018; 2021) and reanalysed surge and sea levels of Muis et al. (2016). However, the performance of DEMs used in these studies was so far only addressed by Bhagabati et al. (2020), who generated a DEM for the Bago River basin based on geodetic data,

supplemented by ALOS data and cross-sectional measurements. When comparing their EnDEM to commonly available SRTM, ASTGTM and HydroSHEDS, they concluded that among these, only HydroSHEDS, which is a hydrologically improved version of SRTM (Lehner et al., 2008), was to some extent comparable to EnDEM (Bhagabati et al., 2020). However, the lack of documentation on vertical reference does not allow for utilising their findings in view of SLR impact assessment.

As a contribution to improved risk assessment for the Ayeyarwady Delta, where the quality of available elevation data is so

far not known, this study aims to assess its elevation in relation to local mean sea level (MSL) by (a) generating a DEM based on geodetic data (called AD-DEM hereafter); (b) using the AD-DEM to evaluate available global DEMs, referenced to local MSL; (c), identifying deltaic areas prone to be effected by RSLR and monsoon flooding in their relation to topography.

Leading research questions that will be addressed are: (i) what is the present elevation of the Ayeyarwady Delta with respect to local MSL?; (ii) what discrepancies exist between local elevation data and available global DEMs?; (iii) which consequences

do different elevation models have when used to assess future RSLR and delta flooding impacts?; (iv) which is – in the absence of ground truthing possibilities – the most trustworthy elevation model when studying RSLR and delta flooding in the Ayeyarwady Delta?; (v) how do these implications call for future coping with SLR?

## 2 Study area

Originating from the confluence of rivers sourced in the southeastern Himalaya, Shan Highlands, Rakhine Mountains, and

Central Dry Zone in Myanmar, the Ayeyarwady River forms a north-southwards oriented fluvial system with a drainage basin covering more than 414,000 km$^2$ (Woodroffe et al., 2006), thereby draining ca. 60 % of the country's territory and supplying water to 90 %  of the Myanmar population (Chen et al., 2020). From the delta apex close to Myanaung north of Hinthada (Fig.



1), the Ayeyarwady first branches (from west to east) into its two first order distributaries Pathein and Myitmakha Rivers
(further downstream entering the Yangon River), and later into the second order distributaries Wakema, Ya Zu Daing, Toe
and Pan Hlaing Rivers (Chen et al., 2020) and several smaller river channels. Spanning over an area of ca. 35,000 km², the
Ayeyarwady Delta is home to more than 15 million people (nearly 30 % of the Myanmar population; Rose et al., 2021),
provides fertile alluvial soils used for agriculture and increasingly for aquaculture (Frenken, 2012; Torbick et al., 2017; Vogel
et al., 2022). Furthermore, the deltaic marshland as well as intertidal flats and islands host several endangered species and
highlight the ecological value of the landscape to be protected in form of key biodiversity areas (Webb et al., 2014; Zöckler
and Kottelat, 2017). However, due to increasing population and human impact, which is also expected to continue, perhaps
grow in the future, this biodiversity as well as the environmental stability and functioning of the delta is threatened (Webb et
al., 2014; Zöckler and Kottelat, 2017; Vogel et al., 2022).

Characterised by the tropical monsoon climate, annual precipitation amounts to 2000–3000 mm in the delta and up to 6000
mm in the mountain ranges (Win Win Zin and Rutten, 2017), with 70–90 % falling in the rainy season (Furuichi et al., 2009;
Zin Mie Mie Sein et al., 2021). In addition to monsoon precipitation, the region is prone to tropical cyclones that may strike
the coast between May and October (e.g., Wang et al., 2013; Brill et al., 2020), causing heavy rainfalls and severe flooding
(Brakenridge et al., 2017). As such, the most prominent and devastating disaster was category 4 tropical cyclone Nargis that
hit the delta in 2008, causing a storm surge of 5 m and storm waves of 2 m height (Fritz et al., 2009). Its unexpected movement,
intensity, and timing with respect to tidal constellation led to flooding up to 50 km inland causing more than 138,000 fatalities
(Fritz et al., 2009). After Nargis, severe flooding occurred due to exceptionally intense monsoon precipitation in 2015, affecting
more than 1.6 million people (with 132 deaths), and up to 530,000 ha of agricultural and aquacultural production areas
(Brakenridge et al., 2017; IFRC, 2017; Government of the Union of Myanmar, 2015).

The Ayeyarwady Delta is a mixed wave and tide dominated system, with particularly the western delta part exposed to waves
(Anthony et al., 2019). Tidal range increases towards the Gulf of Mottama in the east, where it may peak at maximum 7 m
(Rao et al., 2005). Tidal influence varies spatially and temporally. In the most western delta sector, spring and neap tides are
2.2 m and 1.8 m, in the eastern part they are 5.7 and 4.0 m (Kravtsova et al., 2009), while the distributary channels show a
mesotidal range of 2–4 m (Ramaswamy et al., 2004; Chen et al., 2020). Together with spring tides reaching up to 300 km
inland to the delta apex (Hedley et al., 2010), tide dynamics and tidal constellation may significantly contribute to amplify the
impact of storm surge flooding and SLR.






**Fig. 1.** Local elevation model for the Ayeyarwady Delta in Myanmar (a) based on the interpolation of digitised spot and contour heights from topographic maps (areas containing interpolation artefacts are hashed). Spot heights in the north of the delta are irregularly distributed and do not provide full spatial coverage (b). In these parts, topographic information is only provided by contour lines along the river banks (marked with black arrows; c). The spot heights (marked with black circles; d) provide full spatial coverage and a more regular pattern in the southern, central part of the delta. Names and boundaries of administrative districts are shown in a). b) is based on Esri World Imagery (2017); c) and d) are based on topographical map sheets 4143-4 (c) and 4040-1 (d) from East View Geospatial, Inc. (2014).

For the first decade of the 21$^{st}$ century, RSLR has been estimated to 3.4 mm yr$^{-1}$ (Syvitski et al., 2009) in the Ayeyarwady Delta but these rates are solely based on discontinuous tide gauge data between 1916 and 1962 (PSMSL, 2021). By including



a compaction rate of 2–3 mm yr$^{-1}$, Syvitski et al. (2009) postulate up to 6 mm yr$^{-1}$ RSLR for the delta. These rates already exceed the global mean (Church et al., 2006; Masson-Delmotte et al., 2021), thus underpinning its status as "Delta in peril" (Syvitski et al., 2009). Considering global and regional influences, other than land subsidence, sea level is projected to rise by 14–56 cm until the 2050s (based on the reference period 2000–2004), and 21–121 cm until the 2080s (Fee et al., 2017; Horton et al., 2017). Projections of the latest IPCC report (Masson-Delmotte et al., 2021) point to future regional sea levels in 2100 up to 96 cm (intermediate greenhouse gas (GHG) emission scenario SSP2-4.5, 83$^{rd}$ percentile) and 123 cm (high GHG emission scenario SSP5-8.5, 83$^{rd}$ percentile) higher than compared to the period of 1995–2014 (Fox-Kemper et al., 2021; Garner et al., in prep.; Garner et al., 2021). According to the same scenarios, sea level in the Ayeyarwady Delta will have risen by further 67 cm and 98 cm in 2150, respectively (see supplementary material). Since already 0.5 m of the total SLR would lead to a coastline retreat of 10 km (Department of Meteorology and Hydrology and Ministry of Transport, 2012; van Driel and Nauta, 2014), inundation areas are expected to increase, both permanently along the actual coastline due to the flat and low lying topography (Fig. 1) as well as episodically inland due to high tide flooding and storm surges reaching further inland (Fee et al., 2017). As flood frequency and magnitude will increase in all coastal districts, so will saltwater intrusion, thus exacerbating the risk of freshwater shortage, especially during a lengthening dry season (Fee et al., 2017), with severe consequences for agricultural production and food security (e.g., Bucx et al., 2014; Schneider and Asch, 2019; Sakai et al., 2021). However, given that the role of topography and subsidence in the Ayeyarwady Delta has been studied only for specific locations so far, e.g., by observing subsidence at rates of 10–20 mm yr$^{-1}$ in Yangon City in the eastern part of the delta (van der Horst et al., 2018), previous estimates on both RSLR that considered lower values of subsidence or even excluded it (Horton et al., 2017), and flood risk assessments may underestimate future RSLR (e.g., ICHARM et al., 2016; Hein Min Htet, 2017; Liu et al., 2017; Kittiphong Phongsaphan et al., 2019).

## 3 Material and methods

### 3.1 Generation of an elevation model for the Ayeyarwady Delta based on geodetic data

To investigate land elevation in the Ayeyarwady Delta independently of data derived from satellite based measurements, we used height information from recent topographic maps at scale of 1:50,000, which were made and published by the Survey Department, Ministry of Forestry in the Union of Myanmar, in cooperation with the Japan International Cooperation Agency (JICA), and sold by East View Geospatial, Inc. (2014). Map sheets covering the Ayeyarwady Delta were accessed via the Specialised Information Service Cartography and Geodata (SIS Maps) of the Map Department of the *Staatsbibliothek zu Berlin* funded by the German Research Foundation (SIS Maps, 2021). To create a local DEM for our region of interest (ROI; the spatial extent of the Ayeyarwady Delta as defined by Tessler et al. (2015)), 102 topographical map sheets were georeferenced and digitised, covering also the wider surroundings (15° 30' N to 18° 30' N and 94° 0' E to 97° 0' E) to enhance the performance of DEM interpolation. In total, 9179 elevation points were digitised (5673 lying within the ROI). Point elevations are regularly distributed in the central to southern delta parts, while they are more unsystematically and less densely distributed





especially in the northern delta parts. In areas, that are not spatially covered by spot height data, contour lines were digitised.

From these, points were extracted at 250 m and 2000 m intervals, respectively, to supplement the input elevation data for DEM interpolations and testing the impact of considering contour data in DEM interpolation. Furthermore, to reduce the potential of height overestimations in the immediate surroundings of outcrops and improve the quality of the DEM in low lying coastal areas, which are of particular interest in this study, values of elevation higher than a specified threshold are excluded, thereby eliminating outcrops. All data were projected to the Myanmar 2000 datum (JICA et al., 2004).

Interpolation was conducted using the Geostatistical Wizard within the Analysis environment of ArcGIS Pro 2.9.1. Minderhoud et al. (2019) compared 22 interpolation methods for the creation of a DEM in a similar setting, i.e., the Mekong Delta, and found Empirical Bayesian Kriging with empirical transformation and exponential modelling providing the most accurate results (i.e., in terms of absolute accuracy). Thus, similar set-up settings were adopted in this study. The grid cell resolution was defined as justified by the point density following Aguilar et al. (2006). In total, nine versions of AD-DEM

were generated – (i–iii) including digitised spot heights, (iv–vi) including spot heights and contour heights extracted every 250 m, and (vii–ix) including spot heights and contour heights extracted every 2000 m. All of them apply also specific elevation thresholds of 20 m and 10 m, respectively (Table S2). Areas – such as outcrops and northernmost delta parts – that are in reality higher than the elevation threshold applied for DEM interpolation were masked. As data collection for generating the topographic maps used for the AD-DEM was conducted between 2002 and 2004, the location of river channels is likely best

reflected by the SRTM water body mask created in 2000 as changes in geomorphology following river meandering and channel migration will be less compared to water masks of DEMs generated at a later point in time. Therefore, we used the SRTM water body mask file (NASA JPL, 2013) to exclude all areas marked as water from the final DEMs.

**3.2 Processing of global DEMs and determination of local mean sea level (MSL)**

In total, 10 global DEMs were used to assess their performance in the Ayeyarwady Delta. We integrated SRTM, ACE2,

ASTGTM v003, AW3D30, TanDEM-X (30 m), TanDEM-X (12 m), Copernicus DEM, FABDEM, CoastalDEM v2.1, and GLL-DTM v1. An overview of main characteristics such as coverage, resolution, and accuracy, information on eventual edits that have been made after DEM generation as well as a brief description of each DEM is available in the supplementary material. Though elevation data from surface and terrain models may show some systematic offset due to the inclusion of canopy and building heights in the digital surface models (DSMs) compared to real ground elevation assessment of digital

terrain models (DTMs), we include both types of DEMs into the comparison as these DEMs are the most often used, sometimes without considering the DEM type and suitability for its application. Thus, we want to demonstrate the relevance of careful DEM selection because the outcomes of any geomorphological and hydrological study as well as risk assessment and flood modelling will strongly depend on the underlying elevation data (e.g., Wechsler, 2007; Siart et al., 2009; Gesch et al., 2018; Brosens et al., 2022).

Pre-processing of global DEMs and their respective water mask files included mosaicking of single DEM tiles, re-projection to UTM 46 N based on Everest 1830 ellipsoid, clipping to the ROI, and referencing to MSL. We used the latest Mean Dynamic



Topography (MDT) data, i.e., the CNES-CLS18 dataset of Mulet et al. (2021), accessed via the AVISO+ website (https://www.aviso.altimetry.fr/en/data/products/auxiliary-products/mdt/mdt-global-cnes-cls18.html) as an estimation of sea surface height in relation to geoid (see supplementary material for details on local MSL in Myanmar and its potential offset to

a geoid reference). Furthermore, the MDT may provide a more steady and reliable reference than the sparse and discontinuous tide gauge data as tidal range and mean sea level along the delta coast vary in space and time. The variety of vertical datums used by the array of DEMs together with the complex geoid setting along the Myanmar coast made datum transformation to MSL challenging and relations between vertical references of MDT and DEM data needed to be considered when converting from geoid to MSL (Fig. 2).


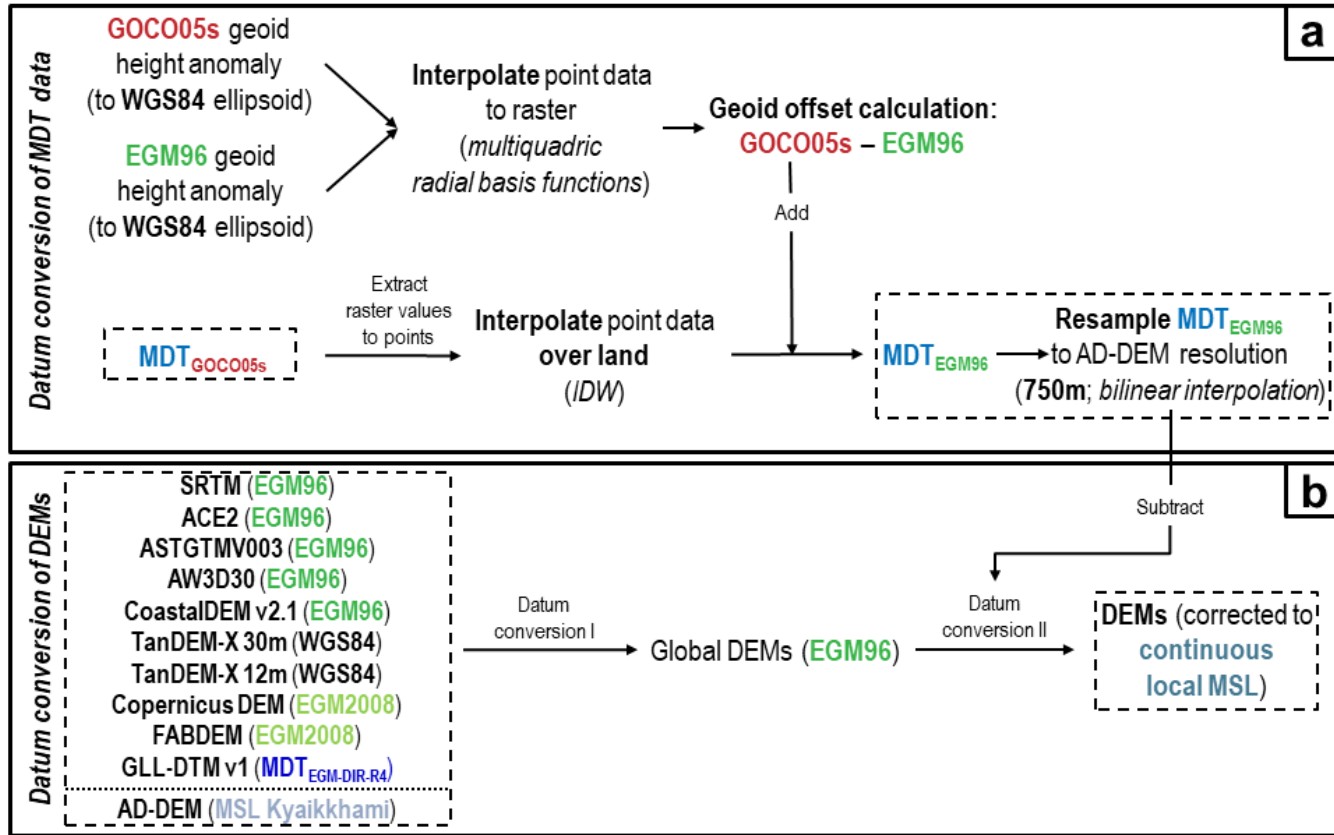

**Fig. 2.** Workflow of datum conversions conducted in this study to reference the Mean Dynamic Topography (i.e., $MDT_{GOCO05s}$) and digital elevation models to a common reference geoid first (i.e., to the EGM96 geoid ($MDT_{EGM96}$ in (a)); datum conversion I in (b)). Subsequently, the corrected MDT data ($MDT_{EGM96}$) was applied to reference all DEMs to the continuous local mean

sea level (MSL) of the Myanmar coast (datum conversion II in b). AD-DEM is the newly developed digital elevation model for the Ayeyarwady Delta based on local data presented in this study.





Processing involved in total three datum conversions. First, given that the majority of DEMs is referenced to the EGM96 geoid, the other datasets were transformed to EGM96 as well. Second, the original MDT data (called $MDT_{GOCO05s}$) was transposed

to EGM96 (called $MDT_{EGM96}$). Subsequently, the third datum conversion was conducted by applying the $MDT_{EGM96}$ to correct the DEMs to MSL. An overview of all datum transformation and processing steps is given in Fig. 2 and all details are provided in the supplementary material.

It should be noted that since the GLL-DTM v1 of Vernimmen et al. (2020) was already referenced to MSL, this DEM did not require the processing as outlined in Fig. 2. However, the GLL-DTM v1 refers to MSL based on an older MDT dataset (Rio

et al., 2014), which is referenced to the EGM-DIR-R4 geoid, where the potential offset between this geoid and the original geoid reference of DEM data was not considered. Due to the missing integration of geoid offset, the GLL-DTM v1 should be used with caution. Although we are unable to correct this ourselves, we included it into our investigations of the Ayeyarwady Delta as it was generated especially for coastal lowlands and to highlight the importance of carefully dealing with vertical datums of datasets using different geoid models.

After referencing the DEMs to MSL, inland water bodies were masked using the DEM-specific water mask files. In addition, cell values with an elevation of more than 7 m below MSL were removed as they likely constitute erroneous height estimates, following Vernimmen et al. (2020). Depending on the type of validation, i.e., with spot heights from topographic maps (East View Geospatial, Inc., 2014) or the AD-DEM, the global DEMs were further processed. The spatial resolution of the DEMs was resampled to match the horizontal accuracy of the AD-DEM using the bilinear resampling technique. For validating the

GLL-DTM v1, the AD-DEM was resampled to its resolution of 0.05° (Table S1). Since outcrops and elevation values outside critical lowlands have been excluded to improve the interpolation of the AD-DEM, these areas were also excluded in the global DEMs to ensure comparability among all DEMs of this study.

Given the large number of DEMs included in this study, together with the most used DEMs in the context of studies with focus on Myanmar as well as the expected similarity between TanDEM-X data and the limited comparability of the GLL-DTM v1,

we decided to exclude the AW3D30, the TanDEM-X 12 m, and the GLL-DTM v1 from the results in the main text. However, results for these DEMs are provided in the supplementary material.

### 3.3 Mapping of areas prone to monsoon flooding

Besides direct validation using spot heights and the AD-DEM, we also performed an additional validation of relative elevation for the DEMs by their ability to match observed floods in deltaic areas affected by inland flooding. We adopted the principal

approach from Minderhoud et al. (2019) who used the correlation of land elevation and tidal flooding to assess the quality of DEMs indirectly. Here, we use mappings of recent monsoon floods due to increased discharge as this allows for investigating the relation between topography and flood pattern also inland. Therefore, we mapped areas inundated during the monsoon periods of 2015 and 2020, respectively, using an image rationing change detection approach on Sentinel 1 imagery (10 m spatial resolution) in the environment of Google Earth Engine (GEE; Gorelick et al., 2017). The pre-event imagery was

recorded in February during the dry season while the selection of imagery capturing maximum flood extents was made based





on investigations of rainfall amounts and river discharge. Monthly accumulated precipitation estimates were obtained from the GPM_3IMERGM v6 product of NASA Global Precipitation Measurement (GPM) program (Huffman et al., 2019) for the rainy months from May to October via the NASA Giovanni web environment (NASA, 2021). Discharge estimates from satellite microwave radiometry were accessed via the River and Reservoir Watch Version 4.5 processor of the Dartmouth

Flood Observatory (Brakenridge et al., 2022) and were downloaded for River Watch stations 25, 29, and 30. Since an array of GEE scripts has been published in the recent past (e.g., Inman and Lyons, 2020; Moharrami et al., 2021; Tripathy and Malladi, 2022), we adopted the script of United Nations Platform for Space-based Information for Disaster Management and Emergency Response (UN-SPIDER, 2019) and slightly adjusted it, i.e., by setting the difference threshold to 1.3, thereby minimising false positive and false negative detections. For further handling and analysis, the data was vectorised in the ArcGIS Pro

environment.

**3.4 Mapping of areas prone to future RSLR**

To gain insights into the potential impact of RSLR in the Ayeyarwady Delta, we estimated areas and population at risk of future RSLR. We calculated areas below certain elevation levels to simulate the impact of selected SLR scenarios by using median projections for Yangon from the NASA Sea Level Projection Tool of the IPCC 6th Assessment Report (Fox-Kemper

et al., 2021; Garner et al., in prep.; Garner et al., 2021). Estimates of 50th and 83rd percentiles from the intermediate SSP2-4.5, the medium to high SSP3-7.0, and the high SSP5-8.5 reference scenarios were used to illustrate the area drowned due to SLR by 2100 and 2150, respectively. From these, only the medium confidence projections were included, thereby excluding the highly uncertain behaviour of the Greenland and Antarctic ice sheets and their contribution to SLR which are reflected by the low confidence projections. To calculate the area below sea level, DEMs were used in their original resolution. The results

were intersected with population data from the LandScan Global 2020 dataset (30 arc sec spatial resolution; Rose et al., 2021). To make our outcomes for decision makers as clear and practicable as possible, results are presented at administrative district level.

**4 Results and interpretation**

**4.1 Performance and accuracy of the AD-DEM**

The new, local AD-DEM not only reveals new insights into the topography of the Ayeyarwady Delta but also constitutes an independent source for evaluating the performance of global DEMs as it is based on field surveys and aerial photography. Given the uneven distribution of spot heights in the delta, point density ranges from 0 to 2.85 points/km$^2$, with the highest densities in Yangon City. While elevation points in townships in the southernmost delta are regularly distributed at densities of up to 0.2 points/km$^2$, their distribution is highly irregular towards townships in the central and northern parts (~0.05–0.1

points/km$^2$), with profound data gaps at 17.2–17.5° N and 17.7–18.0° N. Based on 4,792 spot heights located within the ROI, point density within the delta is ~0.14 points/km$^2$. If areas without nearly any elevation points are excluded, the deltaic area is



28,801 km$^2$ instead of 33,357 km$^2$. Nevertheless, point density does not substantially improve (~0.17 points/km$^2$). This resulted in an optimal spatial resolution of our AD-DEM of 750 m × 750 m (Fig. 1; Tables S1 and S2) as a coarser resolution means a downgrade of the best achievable accuracy, while a finer resolution is not justified by the spatially heterogenous elevation point density.

Among all versions of the AD-DEM, version VI gives the best results with a vertical accuracy of 1.004 m root mean square error (RMSE) and an average standard error of 0.864 m (Table S2). Therefore, we base our investigations of deltaic topography and the comparison with global DEMs on this version of the AD-DEM. Elevation of the delta plain ranges between -1.7 to 8.8 m, with 50 % below 1.3 m (Figs. 1 and S4; Table S3). While northern deltaic areas are more than 8 m above MSL, the AD-DEM indicates a sharp elevation change of up to 10 m at ~17.37° N with elevations even below -1 m above MSL in the northeastern delta. As contour data along the river banks are nearly the only existing data in these areas, their dominance in DEM interpolation may introduce this imperfection (Fig. 1). From the central delta downstream (i.e., ca. 150 km downstream of the apex; Fig. 5a), topography gradually smoothens to overall maximum elevations of several decimetres to 1 m (Figs. 1 and 5a). At 16.25° N, where a second elevation offset of ~50 cm is visible in the AD-DEM at the boundary between irregularly to regularly distributed spot height data (Fig. 1), mean elevation is around 1 m, locally higher in the upper centre of the lowest delta plain and lower at the island of Meinmahla Kyun Wildlife Sanctuary and along the delta shoreline (Fig. 1).

## 4.2 Performance and accuracy of global satellite based DEMs

### 4.2.1 Performance of global DEMs based on visual inspection

How differently the DEMs represent land elevation of the Ayeyarwady Delta becomes already clear by visual inspection (Figs. 3 and S3). All DEMs indicate elevations of more than 10 m for the northernmost delta near the apex and lowest elevations in the southern and southeastern parts, i.e., in Labutta and Pyapon districts. The most striking feature towards the coast is the island of Meinmahla Kyun Wildlife Sanctuary where all DSMs indicate higher elevation compared to the surroundings (Fig. 3). For the entire delta plain, SRTM, ASTGTM v003, and AW3D30 document average heights of ~4.36 to 6.99 m (mean) and 3.68 to 6.32 m (median; Table S3). Among these, height distributions of SRTM and AW3D30 are quite similar while ASTGTM v003 reveals increased elevation counts for classes > 4 m, showing deltawide highest elevations for the entire dataset (Figs. 3 and S4). Both SRTM and AW3D30 suffer from remote sensing artefacts as SW–NE oriented stripes are visible in the SRTM, whereas topography as represented by AW3D30 is affected by distortions that reflect the sensor swaths (Figs. 3 and S3). With average heights of ~1.68 m (mean) and ~0.94 m (median; Table S3), ACE2 indicates the lowest elevation for the delta plain, which is reflected also by the comparably high frequency of counts below MSL (Fig. S4). As ACE2 is the product of merging SRTM with altimetry data, it suffers from the same stripe artefacts as SRTM. In contrast, artefacts are not visible in TanDEM-X DEMs and processed versions like Copernicus DEM and FABDEM. As the TanDEM-X 30 m was generated by averaging heights from TanDEM-X 12 m data, both their statistics and rasters are very similar. However, maximum heights of more than 400 m above MSL point towards the presence of unrealistic elevation values that are likely erroneous as the delta ROI does





not include any mountains or elevated terrain (Table S3). Consequently, these values have been corrected in Copernicus DEM

and FABDEM. Removing trees and buildings from the Copernicus DEM makes the FABDEM delivering a smoother surface
with less deviation of elevation data and lower average heights of ~2.75 m (mean) and ~1.83 m (median; Table S3). Though
the TanDEM-X, Copernicus DEM, FABDEM, and CoastalDEM v2.1 share the general pattern of height distributions, lowest
elevations of < 2 m are especially frequent in the CoastalDEM v2.1 (Fig. S4). Mean and median elevation of the CoastalDEM
v2.1 nearly match the AD-DEM (Table S3), however, visual investigation reveals that the same sensing artefacts as observed

for SRTM are present. Hence, this implies that – since the CoastalDEM v2.1 used elevation data from the NASADEM, which
is an improved SRTM – both processing and corrections of SRTM applied in NASADEM and CoastalDEM v2.1, respectively,
could not account for these artefacts adequately. According to the GLL-DTM v1, average heights are higher than observed for
the CoastalDEM v2.1 and more similar to FABDEM and Copernicus DEM while there is much less scatter of elevation counts
(Table S3; Fig. 3). The latter may be attributed to its comparably coarse resolution that does not capture small scale topography.

However, given that the GLL-DTM v1 refers to a different MDT dataset than the one used in this study, without being corrected
for this unknown geoid offset, it remains open whether this vertical bias is responsible for indicating higher elevations than,
e.g., AD-DEM (Table S3; Figs. S3 and S4).





**Fig. 3.** The new local AD-DEM (a) in comparison to global digital elevation models in the Ayeyarwady Delta (b–h).



### 4.2.2 Direct validation with spot heights from topographic maps

After extracting all 4,792 spot heights inside the ROI from the DEMs using bilinear interpolation of values at point location, we included only those containing data in the specific DEM, respectively. Spot heights range from -2.31 m to 214.99 m above MSL with mean and median elevation of 6.06 m and 1.69 m (Fig. S5). Largest inaccuracies are documented for the ASTGTM

v003 with average height residuals of 2.75 m (mean) and 4.38 m (median), as well as 6.21 m absolute error and 9.20 m RMSE (Tables 1 and S4; Figs. S6 and S7). Thus, $R^2$ of 0.75 indicates the lowest quality compared to other DEMs of this study (Tables 1 and S4). Though SRTM performs better than ASTGTM v003 in terms of height residuals, it still indicates elevations overall higher than the spot heights. In contrast, the corrected SRTM version ACE2 shows mainly negative deviations and implies a good correlation with $R^2 = 0.86$ (Table 1; Fig. S6). However, the absolute error is still more than 3 m, and the elevation data

shows substantial scatter as reflected by the maximum standard deviation σ compared to all DEMs. TanDEM-X and improved versions like Copernicus DEM and FABDEM reveal not only lower height residuals but also show point densities that are considerably higher than for the others (Figs. S6–S9). Nevertheless, RMSE is in similar range as for SRTM and ACE2, being with 7.23 m minimal for the Copernicus DEM (Table 1). The recently published FABDEM shows the lowest absolute error of 2.73 m, median height residual of ~0.24 m, and it performs nearly as well as ACE2 in terms of $R^2$ (Table 1). Being generated

for coastal regions in particular, the CoastalDEM v2.1 indicates even less uncertainty with median height residuals on centimetre scale and median elevation differs only 0.12 m from median spot height elevation (Table 1; Fig. S7). In contrast, mean and absolute errors are increased, and RMSE is the second largest amounting to 7.81 m. $R^2$ is slightly lower than for Copernicus DEM and FABDEM (Table 1). Overall, all DEMs indicate significantly lower elevations for the mountain ranges of the Rakhine Mountains, and so far, we have no explanation for this.


**Table 1.** Statistics for DEMs minus spot height elevation at spot height location. Best performances are shown in italics. DEMs were used in their original resolution and vertically referenced to the same mean sea level. N – Number of spot heights in the ROI, with no data values excluded for each DEM respectively; HR – Height residual; MAE – Mean absolute error; RMSE – Root mean square error; Min – minimum DEM elevation at spot height location; Max – maximum DEM elevation at spot

height location; Mean spot – mean elevation of all spot heights included in the comparison; Mean DEM – mean DEM elevation at all spot height locations included in the comparison; Median spot – median elevation of all spot heights included in the comparison; Median DEM – median DEM elevation at all spot height locations included in the comparison; σ spot – standard deviation of elevation of all spot heights included in the comparison; σ DEM – standard deviation of DEM elevation at all spot height locations included in the comparison; σ HR – standard deviation of height residuals; $R^2$ – Coefficient of determination.

| DEM | N | Mean HR | MAE | RMSE | Min DEM | Max DEM | Mean spot | Mean DEM | Median spot | Median DEM | Median HR | σ spot | σ DEM | σ HR | $R^2$ |
|---|---|---|---|---|---|---|---|---|---|---|---|---|---|---|---|
| SRTM | 4781 | 1.17 | 4.05 | 7.36 | -3.31 | 182.88 | 6.00 | 7.17 | 1.68 | 4.20 | 2.09 | 16.11 | 11.90 | 7.26 | 0.83 |
| ACE2 | 4358 | -1.87 | 3.32 | 7.33 | -6.41 | 189.09 | 6.39 | 4.52 | 1.77 | 1.32 | -0.80 | 16.78 | 12.44 | 7.08 | *0.86* |



| | N | Mean HR | MAE | RMSE | Min DEM | Max DEM | Mean spot | Mean DEM | Median spot | Median DEM | Median HR | σ spot | σ DEM | σ HR | R² |
|---|---|---|---|---|---|---|---|---|---|---|---|---|---|---|---|
| ASTGTM v003 | 4776 | 2.75 | 6.21 | 9.20 | -2.31 | 165.07 | 6.00 | 8.75 | 1.69 | 6.82 | 4.38 | 16.11 | 10.49 | 8.77 | 0.75 |
| TanDEM-X 30 m | 4678 | *-0.23* | 3.09 | 7.52 | -4.97 | 166.03 | 6.09 | 5.86 | 1.71 | 2.42 | 0.54 | 16.27 | 11.63 | 7.52 | 0.82 |
| Copernicus DEM | 4776 | -0.52 | 2.90 | *7.23* | -1.12 | 166.11 | 6.00 | 5.48 | 1.69 | 2.25 | 0.41 | 16.11 | 11.53 | 7.21 | 0.84 |
| FABDEM | 4776 | -1.01 | *2.73* | 7.40 | -1.12 | 160.34 | 6.00 | 4.99 | 1.69 | 2.04 | 0.24 | 16.11 | 10.96 | 7.34 | 0.85 |
| CoastalDEM v2.1 | 4746 | -1.65 | 2.94 | 7.81 | -2.28 | 192.21 | 6.00 | 4.34 | 1.70 | 1.58 | *-0.06* | 16.10 | 10.81 | 7.63 | 0.83 |


To assess the quality of the DEMs for the low lying delta parts, we validated them against spot height elevations less than 10 m above MSL (Tables 2 and S5). Integrated spot heights (Σ 4,112) range from -2.31 m to 9.97 m above MSL. As observed for the DEM comparison with all spot heights, similar trends are found for low elevations, with ASTGTM v003 being the least accurate one (Table 2). SRTM and AW3D30 are characterised by comparably less deviations; ACE2 and TanDEM-X perform

better, showing even lower height residuals and higher point densities (Figs. S10 and S11; Tables 2 and S5). However, they all show RMSE in the same range of 2.58 to 3.87 m and overall low quality with $R^2$ of 0.28. Best results are delivered by those DEMs that have been corrected by processing with additional datasets. While the CoastalDEM v2.1 shows the best accuracy with average residuals of 0.04 m (mean) and 0.14 m (median), as well as 1.31 m of absolute error and 1.79 m RMSE, the FABDEM documents the highest level of agreement with spot height data ($R^2$ = 0.45; Table 2), which is also reflected by

maximum point densities (Fig. S11). In contrast, the relatively poor performance of the GLL-DTM v1 in terms of $R^2$ likely results from its spatial resolution being too coarse to give adequate results for the spot height comparison, which adds to the unknown bias of its different vertical datum (Table S5).

**Table 2.** Statistics for DEMs minus spot height elevation at spot height location for the low lying delta plain (spot heights <

10 m above MSL). DEMs were used in their original resolution and vertically referenced to the same mean sea level.

| | N | Mean HR | MAE | RMSE | Min DEM | Max DEM | Mean spot | Mean DEM | Median spot | Median DEM | Median HR | σ spot | σ DEM | σ HR | R² |
|---|---|---|---|---|---|---|---|---|---|---|---|---|---|---|---|---|
| SRTM | 4105 | 2.50 | 3.01 | 3.87 | -3.31 | 34.07 | 1.82 | 4.32 | 1.33 | 3.75 | 2.31 | 2.20 | 3.28 | 2.95 | 0.23 |
| ACE2 | 3694 | -0.42 | 1.92 | 2.58 | -6.41 | 19.13 | 1.87 | 1.45 | 1.35 | 0.85 | -0.56 | 2.26 | 2.87 | 2.55 | 0.28 |
| ASTGTM v003 | 4100 | 4.82 | 5.20 | 6.08 | -2.31 | 33.91 | 1.82 | 6.65 | 1.33 | 6.43 | 4.86 | 2.20 | 3.47 | 3.71 | 0.04 |
| TanDEM-X 30 m | 4002 | 1.14 | 1.78 | 3.01 | -4.97 | 40.66 | 1.83 | 2.97 | 1.34 | 2.04 | 0.68 | 2.21 | 3.22 | 2.79 | 0.28 |
| Copernicus DEM | 4100 | 0.79 | 1.60 | 2.39 | -1.12 | 23.62 | 1.82 | 2.62 | 1.33 | 1.82 | 0.55 | 2.20 | 2.76 | 2.26 | 0.37 |





| | | | | | | | | | | | | | | |
|---|---|---|---|---|---|---|---|---|---|---|---|---|---|---|
| FABDEM | 4100 | 0.46 | 1.32 | 1.84 | -1.12 | 21.27 | 1.82 | 2.28 | 1.33 | 1.67 | 0.40 | 2.20 | 2.21 | *1.79* | *0.45* |
| CoastalDEM v2.1 | 4078 | *0.04* | *1.31* | *1.79* | -2.28 | 17.18 | 1.83 | 1.87 | 1.34 | 1.35 | *0.14* | 2.20 | 2.02 | *1.79* | 0.42 |
| AD-DEM | 3616 | -0.03 | 0.53 | 0.85 | -1.26 | 8.37 | 1.60 | 1.56 | 1.29 | 1.19 | 0.03 | 1.85 | 1.53 | 0.85 | 0.80 |

In the absence of any additional independent ground truth measurements in the Ayeyarwady Delta, we checked the AD-DEM against the spot heights (< 10 m only) used for interpolation to assess the performance of the interpolation algorithm for the AD-DEM. For low lying elevations, the AD-DEM is accurate at 0.53 m absolute error and 0.85m RMSE, showing average

height residuals of -0.03 m (mean) and 0.03 m (median). The quality of correlation is documented by $R^2$ of 0.80 (Table 2). Overall, and as the statistics of this comparison with spot heights show, it turns out that relying on individual, selected accuracy parameters does not provide a holistic picture of the performance of a DEM. Rather, mean and median residuals, absolute errors as well as RMSE and $R^2$, respectively, indicate varying degrees of DEM quality. Therefore, we suggest that the collective of these statistics provides the best answer to which DEM represents the true elevation of the delta best.

**4.2.3 Direct validation of global DEMs with AD-DEM**

Validation with the AD-DEM serving as reference allows to assess and quantify spatial patterns and differences, thereby complementing the comparison of DEMs by visual inspection (Sect. 4.2.1) and at spot height locations (Sect. 4.2.2). By subtracting resampled DEM versions by the AD-DEM, the resulting maps and their statistics document both similarities and differences between the DEMs (Figs. 4 and S12; Table S6). Relative to AD-DEM, in general, most DEMs tend to overestimate

elevation of the delta plain. SRTM, ASTGTM v003, and AW3D30 show deviations of several metres that are on average 4.87 m (mean) and 4.75 m (median) for ASTGTM v003 and 3.83 m (mean) and 2.22 m (median) for SRTM (Figs. 4 and S12; Table S6). Since AW3D30 strongly suffers from sensing artefacts (see Sect. 4.2.1), these become also visible in the difference map (Fig. S12a). While ASTGTM v003 indicates maximal positive deviations from AD-DEM, ACE2 documents the most negative ones, i.e., lowest elevation compared to AD-DEM (Figs. 3 and 4; Table S6). The error statistics of ACE2 range between that

of the aforementioned DEMs and TanDEM-X data which perform moderately compared to all other DEMs of this study (Figs. 4 and S12; Table S6). Height residuals and uncertainties of TanDEM-X are slightly reduced by Copernicus DEM and FABDEM (Fig. 4d–f; Table S6). The FABDEM differs from AD-DEM by 0.63 m (mean) and 0.46 m (median) revealing an uncertainty of 2.00 m RMSE (Fig. 4f; Table S6). Its quality relative to AD-DEM is further described by $R^2$ of 0.53. Beside the FABDEM (and the GLL-DTM), the CoastalDEM v2.1 shows the greatest similarity to AD-DEM as residuals are mainly

present on decimetre scale (Fig. 4g; Table S6). Thus, uncertainty is minimal with RMSE = 1.75 m while correlation to AD-DEM only slightly improves compared to FABDEM, amounting to $R^2$ = 0.55 (Table S6).

Aside from the central and southern delta plain, all DEMs underestimate delta elevation relative to AD-DEM in the northern part, i.e., where mainly contour data (only present along river banks) could be used to interpolate the AD-DEM. Similarly, all DEMs show higher elevations directly south of this zone, overall reflecting the interpolation artefacts in the AD-DEM.


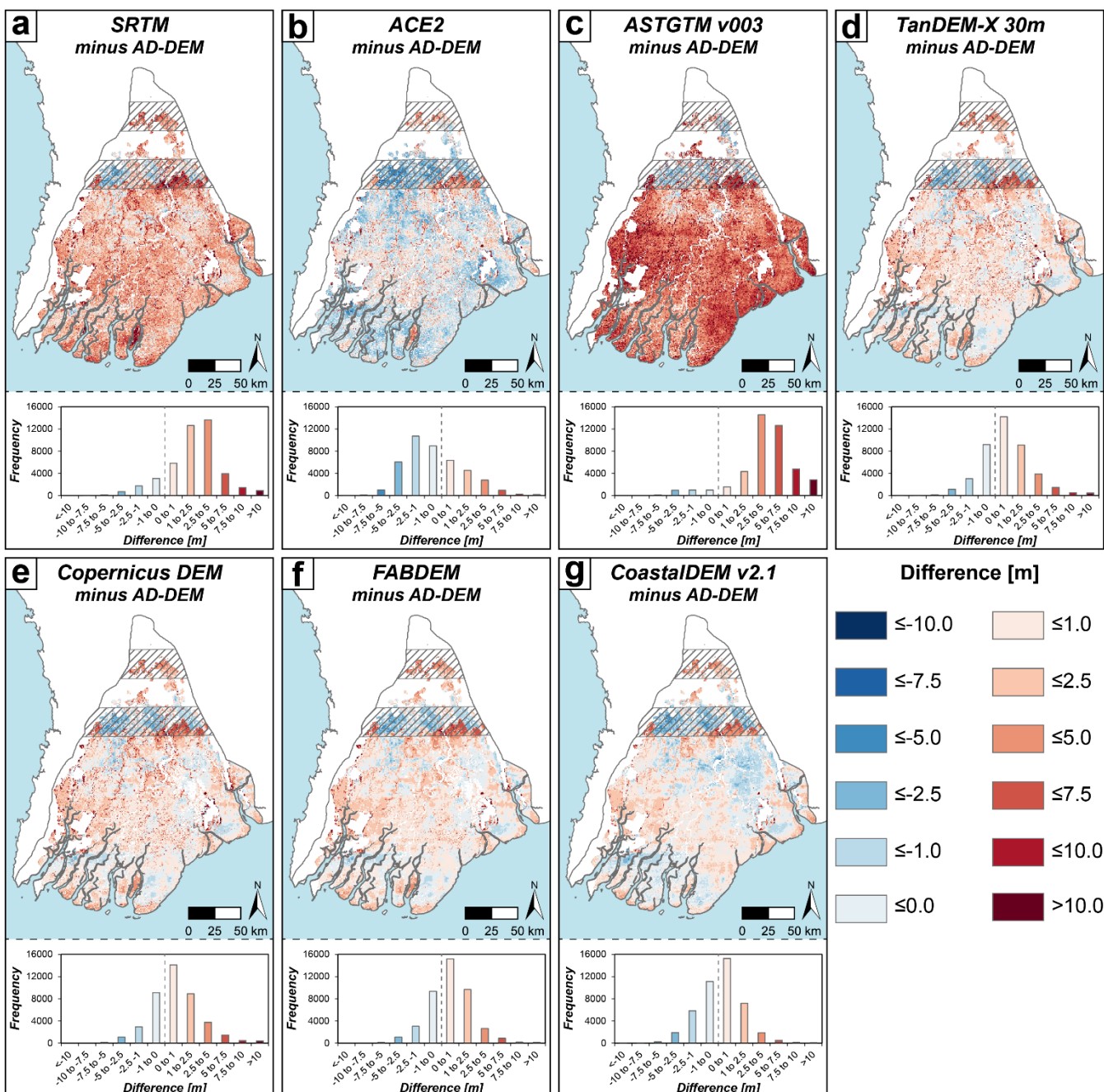

**Fig. 4.** Differences between global DEMs and the AD-DEM together with frequency distributions of height residuals (in m).
Areas where AD-DEM has interpolation artefacts are hashed.



### 4.2.4 Indirect validation of DEMs by comparison with areas drowned during recent flood events

Investigating the topography of areas that have recently been affected by flooding provides an indirect way to detect potential inaccuracies in the DEMs. Though pluvial and fluvial flooding may be less correlated with absolute elevation than, e.g., tidal flooding, we expect a correlation of flooded areas with topographic depressions and other low relief surfaces and use this spatial relation to validate the DEMs in terms of relative elevation. The DEMs already reflect the spatially varying extent of the 2015 and 2020 monsoon floodings by their distributions of elevation counts. The more southward reaching of the 2020 flooding towards low lying delta parts is reflected by higher frequencies of low elevation values in the DEMs (Fig. S17; further details are given in the supplementary material). To evaluate the DEMs, we generated several elevation profiles throughout the delta that cross particularly flood prone areas, thereby including different land surface types as well as typical features of fluvial landscapes (Figs. 5 and S18).

For the upper delta plain, which is characterised by abandoned river channels, levees, shrubland vegetation and cropping (Vogel et al., 2022), most DEMs show similar cross sections for profile B–B' where depressions coincide with mapped inundation (Fig. 5c). However, while ASTGTM v003 indicates elevation changes of sometimes more than 20 m over short distance that are likely unrealistic within an individual inundation area, ACE2 performs partly contrary to the other DEMs, particularly in the western part of profile B (Fig. 5c). Here, according to ACE2, only slopes of a depression would have been affected by flooding while the lowest point would have remained dry. In the most western part of the ROI, profile C–C' documents the transition of the Rakhine Mountains towards the Ayeyarwady basin north of Pathein that is characterised by shrubland and agricultural land use (Vogel et al., 2022). Since most of the DSMs show abrupt elevation changes within individual inundation zones, only corrected DSMs (or nearly DTMs) such as Copernicus DEM, FABDEM, and CoastalDEM v2.1 that show moderate changes seem reasonable (Fig. 5d). In contrast, GLL-DTM v1 and AD-DEM only reflect the general, large depression of Pathein as small scale topographic features are either averaged out or not recorded at all due to data paucity. Lying in the central delta, profile D–D' crosses a sequence of silted up oxbows, which are used for the cultivation of irrigated crops (Vogel et al., 2022), separated by former river banks, covered with shrubs and trees. All TanDEM-X based DEMs indicate a similar relative topographic setting, with differences in absolute elevation becoming particularly obvious along vegetated former river banks. The TanDEM-X based DEMs show major differences from ASTGTM v003 and AW3D30, and minor deviations from SRTM (Fig. 5e). From the high and mid resolution DEMs, CoastalDEM v2.1 indicates the smoothest topographic profile, showing the highest correspondence with AD-DEM and also underlying spot heights (Figs. 5e and S7). Overall, depressions and flat terrains with maximum elevations of 8 m above MSL were the most extensively affected by the monsoon floodings (Fig. 5e). However, given that individual inundation areas are small compared to DEM resolution, this does not allow for further conclusions other than already stated above, where ASTGTM v003, ACE2, and AW3D30 show less correspondence with flood mappings.



**Fig. 5.** Delta elevation along longitudinal (a) and cross sections (c–g) in deltaic areas affected by flooding during the monsoon seasons of 2015 and 2020 (b; Base map: Esri World Imagery, 2017). Elevation profiles are based on DEMs resampled to 750 m spatial resolution. Blue bars on top of the profiles indicate the location of inundated areas along the profiles. The black dashed rectangle in (a) marks the low elevation zone along the longitudinal profile that is also shown enlarged.





At profile E–E' in the eastern delta, characterised by a marshy environment used for the cultivation of irrigated crops (Vogel et al., 2022), flooding is more extensive than for the silted up oxbows, with the largest contiguous inundation areas being located in the flattest topographies (Fig. 5f). Given that embankments in the Ayeyarwady Delta were designed to protect against floods of up to 20-yr recurrence (e.g., Myat Myat Thi et al., 2012), the dike present in the westernmost part of the cross section has a relative elevation of 4–5 m and is therefore best represented by the CoastalDEM v2.1 as most of the other DEMs indicate higher relative dike elevation (Fig. 5f). However, the CoastalDEM v2.1 indicates significantly lower absolute elevations than those of independent elevation data (see also Fig. 4g) and implies an atypical bimodal profile of the dike. Thus, the FABDEM provides the best compromise and best reflects both the relative height of the dike and the smooth topography of the embanked paddy fields. Though the integration of only two events might be insufficient to characterise flood risk in detail, there is an obvious relation between relief and inundation, promoting low relief and water logging conditions inside the embanked area. Finally, profile F–F' crosses the lowest delta plain, once a transition zone towards coastal mangrove marsh, nowadays used for cultivating irrigated and dry crops (Vogel et al., 2022). This area was flooded particularly during the 2020 monsoon season and is characterised by many small individual inundation areas (Fig. 5g). However, these are not restricted to depressions, which are mainly located distal to the distributary channels and in the centre of deltaic islands, but rather occur also in the surroundings, which most DEMs indicate to have a pronounced relief (Fig. 5g). As the profile section does not cross any significantly inclined topographical features like steep river banks or beach ridges, which can be found directly along the shore, the FABDEM, CoastalDEM v2.1, GLL-DTM v1, and AD-DEM likely reflect the topographic setting more realistically than the others, with the FABDEM performing most similar to the AD-DEM (Fig. 5g).

### 4.3 Estimation of coastal area at risk of future sea level rise

Combining the DEMs with the latest datasets of SLR projections and actual population (as of 2020) allows for the first time to quantify both the spatial extent and number of people prone to flooding by SLR in the Ayeyarwady Delta. By 2100, regional sea level will be 0.712 m (50th percentile of SSP24.5 scenario) to 1.229 m (83rd percentile of SSP5-8.5 scenario) higher than in the reference period of 1995 to 2014 (Fox-Kemper et al., 2021). By 2150, which is the maximum of the projection timeline, even delta parts of more than 2.210 m elevation could be drowned (Figs. 6, S19 and S20; Table S7; Fox-Kemper et al., 2021). However, these future sea levels will be reached even sooner when rates of land subsidence are higher or more dynamic than considered in the vertical land motion rates of the IPCC 6th Assessment Report.





**a–h) Area below sea level for relative sea level rise scenarios**

| 0.712 m | 0.820 m | 0.929 m | 0.955 m | 1.086 m | 1.229 m |

**Fig. 6.** Area below future mean sea level according to local and global DEMs following median sea level rise projections for Yangon for 2100 (compared to the baseline period 1995–2014) from the Sea Level Projection Tool of the IPCC 6th Assessment Report (Fox-Kemper et al., 2021; Garner et al., in prep.; Garner et al., 2021). Note that in case of higher RSLR, these scenarios will be reached sooner than 2100.

For the 0.712 m SLR scenario, the range of DEMs indicates in total 18.5 to 11072.0 km² (0.1 to 42.5 %) of the administrative deltaic area to be lost, an area today populated by ~3800 to more than 3 million people (2020 population; Fig. S21; Table S7). With 1.229 m SLR under high SSP5-8.5 scenario, more than 50 % of the delta could fall below sea level, thus affecting an area where ~25 % of the delta's 2020 population is living (Fig. S21; Table S7). However, these numbers can be narrowed down when using the DEMs with the most robust performance in Sect. 4.2. FABDEM, CoastalDEM v2.1, and AD-DEM indicate 7.0 %, 17.5 %, and 31.3 % of land below a 0.712 m raised sea level and even 30.7 %, 42.4 %, and 46.7 % if SLR



reaches 1.229 m (Table S7). Within these areas, today more than 359,000 (FABDEM; 0.712 m SLR scenario) to nearly 3 million people (AD-DEM; 1.229 m SLR scenario) are living. Sea levels higher than 2.210 m compared to the baseline reference 1995–2014, as projected by the IPCC for 2150, will increase the drowned delta area by further ~20–30 %, i.e., to 53.2 % for
the FABDEM, 67.5 % for the CoastalDEM v2.1, and 65.0 % for the AD-DEM (Table S7), while estimates based on lower GHG emission scenarios from 2150 projections are in the range of high GHG emission scenarios for the year 2100. Accordingly, these DEMs indicate a population of 3.6 to 5 million people living today in areas below 2.210 m above MSL; this is equivalent to one third of the total delta population (as of 2020). ACE2 even points to nearly 5.2 million people living at risk of SLR, while in contrast, ASTGTM v003 reveals the lowest estimate of ~64,500 people (Table S7).

Beside those estimates, the DEMs show also different spatial patterns of SLR impact, thereby again reflecting sensing artefacts like stripes as discussed in Sect. 4.2.1 (Figs. 6, S19 and S20). The majority of the DEMs, i.e., all TanDEM-X based DEMs as well as the AD-DEM and GLL-DTM indicate that moderate SLR will largely impact the south-eastern delta while only ACE2, CoastalDEM v2.1, and AD-DEM show also areas affected more inland (Figs. 6, S19 and S20). However, all of them point towards the administrative districts of Pyapon, Labutta, and Myaungmya, which are particularly vulnerable to SLR as the cities
of Pyapon, Dedaye, Bogale, Kyaiklat, Labutta, Mawlamyinegyun, Myaungmya, Einme, and Wakema, each of them with more than 5,000 inhabitants, are located in those areas that will be drowned by up to ~94 % in the future (Figs. 7 and S21–S25). The least affected districts will be Hinthada and Thayarwady in the northernmost delta. Beside FABDEM and CoastalDEM v2.1, which revealed a comparably good performance in comparison with independent elevation data, the SLR impact assessments based on these DEMs lead to considerably different estimates of affected district areas and population, thereby highlighting
the range of uncertainty for each district (Figs. 7 and S24). For the SLR scenarios as projected by the IPCC for 2100, largest spatial uncertainties amount to ~30 % for Myaungmya and ~24 % for Labutta districts, equivalent to an absolute range of up to ~348,000 ± 57,000 and ~487,000 ± 119,000 people (2020 population) who live at risk of 1.229 m SLR in Labutta and Myaungmya, respectively (Fig. S24). Even higher sea levels as are projected for 2150 further turn out the exposure and vulnerability of the densely populated metropolitan area of Yangon and Ma-ubin district with the smaller cities of Ma-ubin,
Pantanaw, Nyaungdon and Danubyu (Figs. 6, 7 and S24–S25). Based on the 2.210 m SLR scenario, between ~322,000 and 682,000 people (2020 population) will be affected in Ma-ubin district, reflecting the largest spatial uncertainty for the best performing DEMs (i.e., ~35 %, equivalent to > 360,000 people; Figs. 7 and S24). Though spatial uncertainty of SLR impact is less for the Yangon districts (i.e., ~8–30 %), the range of affected population increases as population density is magnitudes higher than in other districts (Fig. S25). Thus, based on the comparison of AD-DEM, FABDEM, and CoastalDEM v2.1, we
estimate ~362 ± 265 thousand people in Yangon North, ~317 ± 109 thousand in Yangon South, ~368 ± 93 thousand in Yangon East, and ~30 ± 29 thousand people in Yangon West (2020 population) to be affected by SLR until 2150 (Figs. S23 and S24). However, much larger uncertainties of nearly 95 % exist if considering all DEMs of this study, despite their qualitative performance. As these largest uncertainties are obtained for the district of Pyapon under the projection of 2.210 m SLR, translation into population counts to a range of ~20 to ~974 thousand people at flood risk due to SLR (Fig. S23).






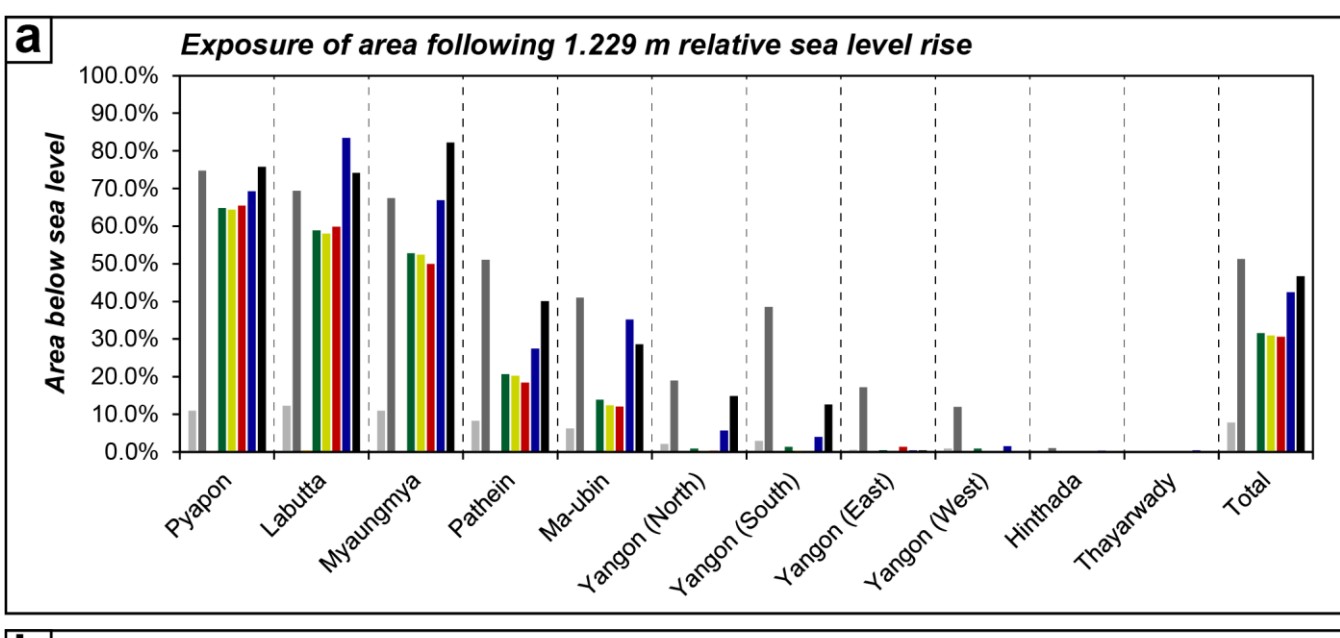

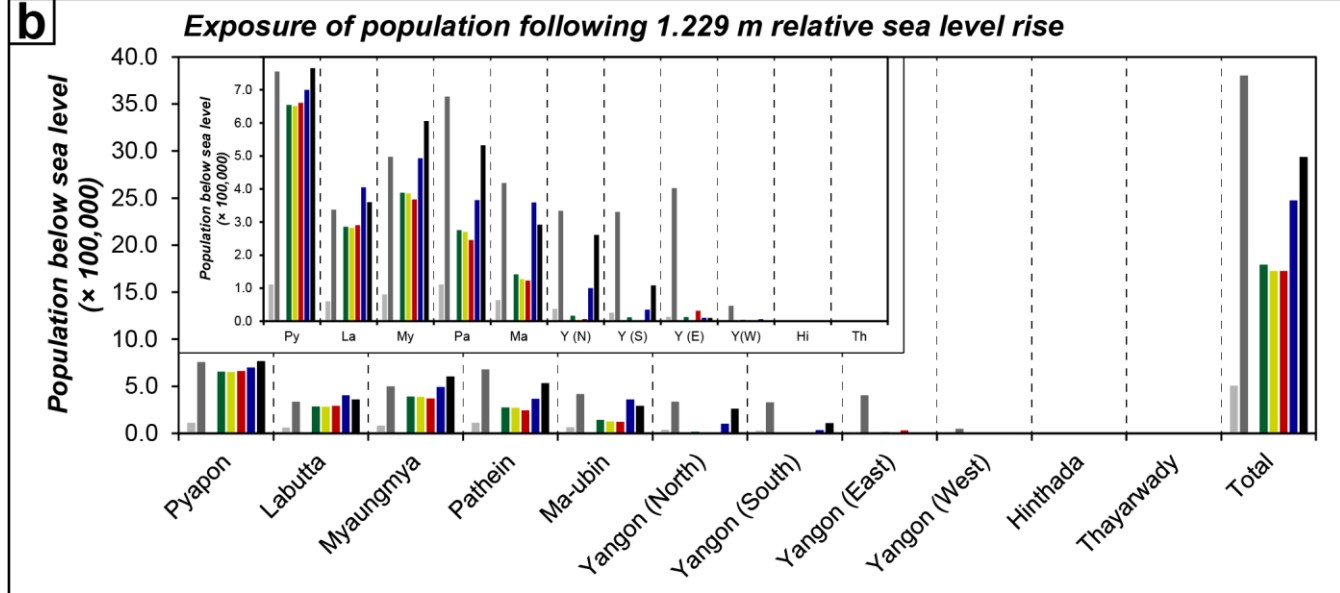

SRTM; ACE2; ASTGTM v003; TanDEM-X 30m; Copernicus DEM; FABDEM; CoastalDEM v2.1; AD-DEM

**Fig. 7.** Area (a) and people (b) affected by sea level rise of 1.229 m as projected by SSP5-8.5 (83[rd] percentile) for 2100 for each administrative district of the Ayeyarwady Delta and the entire delta. Estimates of affected district population are enlarged in the insets. All population estimates are based on the actual population of 2020 (Rose et al., 2021).


Overall, in view of all SLR scenarios considered in this study, the most extensive impact on almost all deltaic districts will occur when SLR exceeds 0.929 and 1.086 m by 2100 as well as 1.939 and 2.210 m by 2150 (Fig. S21), pointing towards the



impending challenge of potential huge out-migration of the delta. As vertical land motion rates of the IPCC 6[th] Assessment Report may not sufficiently capture the contributions of land subsidence and land elevation change, actual RSLR may exceed

these IPCC projections, thereby pointing to the uncertainty in timing of SLR impact.

## 5 Discussion

### 5.1 Suitability of DEMs for studying flood risk in the Ayeyarwady Delta

Based on local geodetic measurements, the here presented, local AD-DEM reveals unique and highly valuable information on land elevation of the Ayeyarwady Delta at high vertical accuracy of 0.86 m mean average standard error and 1.00 m RMSE.

Therefore, it not only serves for evaluating global satellite based DEMs but rather provides new reliable insights into the exposure of the delta to RSLR. Focussing on low elevations of ≤ 10 m makes it particularly suitable to study the central to lower delta plain, which are the most vulnerable to RSLR. Though its spatial resolution of 750 m × 750 m may hamper its suitability on small local scale (e.g., in terms of identifying single topographical features), it proves particularly well for studying larger local/subregional (e.g., at township level) to deltawide patterns of topography, land elevation, and correlated

exposure to floods and RSLR.

To evaluate the performance of DEMs in the Ayeyarwady Delta, we assessed 10 global DEMs, firstly based on their visual impression, secondly, in relation to independent elevation data in terms of spot heights and the AD-DEM, and thirdly, we conducted an indirect validation by analysing the topographies of areas inundated during recent flooding events. While stripes and sensing artefacts constitute obvious qualitative constraints for some DEMs (i.e., SRTM, AW3D30, CoastalDEM v2.1) or

turn out in form of unrealistic elevation values that are likely erroneous (e.g., TanDEM-X), accuracies are specified by an array of error metrics. Overall, none of the DEMs included in this study proves best for each metric and each validation type. Since the datum of the GLL-DTM v1 could not be tied to the same local MSL as all the other DEMs, its statistics calculated in this study may not be reliable. When comparing the DEMs to spot heights (and excluding the GLL-DTM v1), Copernicus DEM shows the smallest RMSE while FABDEM indicates the smallest MAE and median height residual gets minimal for

CoastalDEM v2.1 (Table 1). In contrast, ACE2 shows the highest agreement with the spot height data. However, in view of low elevations (< 10 m), CoastalDEM v2.1 and FABDEM reveal the best accuracies (Table 2). The latter DEMs also show the lowest errors in comparison to AD-DEM (Table S6), where the CoastalDEM v2.1 indicates uncertainties of 0.10 to 0.46 m less than the FABDEM while correlation with AD-DEM improves only slightly (Table S6).

However, the lack of evenly distributed spot heights together with the presence of elevation data in the upper delta plain only

along the river banks constitutes itself a source of uncertainty for the AD-DEM, with the elevation of the delta plain potentially not completely represented locally. Since we observe elevation offsets to spot heights in the Rakhine Mountains for all DEMs exceeding up to -2.5 σ, these are likely systematic biases we have no explanation for, but which may be attributed to inconsistencies during surveying and data collection that may occur over such a large area (i.e., > 44,000 km[2]; JICA et al., 2004). Still, as these biases are restricted to the mountain ranges, we highlight the value of the AD-DEM for assessing land



elevation in the low lying delta because it represents local elevation with a vertical accuracy at decimetre to metre scale, being
independent of satellite based measurements, where errors are in the range of several metres.

Furthermore, the systematic offset between DSMs and DTMs needs to be considered as the AD-DEM constitutes a DTM,
which is also reflected by overall lower elevations in vegetated areas such as in the mangroves of the Meinmahla Kyun Wildlife
Sanctuary or along former point bars now covered by shrubs and trees (Figs. 4, 5, S12 and S18). Rightfully, the corrected

DSMs, namely FABDEM and CoastalDEM v2.1, which are nearly DTMs, show the closest similarity to AD-DEM and also
the most realistic setting in comparison with inundation areas. However, their relation to AD-DEM differs depending on the
local profile setting. While discrepancies are lowest for smooth, flat terrain characterised by low-growing crops and vegetation,
the CoastalDEM v2.1 seems to level out relief changes present between former point bars and oxbows or yields an atypical
profile for a dike whereas the FABDEM reflects a more pronounced and – at least in terms of the dike – more realistic

topography. From the central to the lower delta, absolute elevations of both FABDEM and CoastalDEM v2.1 are mainly
consistent with the AD-DEM (Fig. 4), but with the tendency that the CoastalDEM v2.1 is locally 2–3 m lower than AD-DEM
while the FABDEM is either in line or slightly below AD-DEM in marshy environment cultivated with irrigated and/or dry
crops (Fig. 5f and g).

In view of studying the impact of flooding in the Ayeyarwady Delta, the large scale pattern of area at risk mapped based on

AD-DEM also serves as reference to evaluate the performance of the other DEMs. However, the relatively coarse resolution
of the AD-DEM makes it less suitable to function as benchmark to evaluate the small scale spatial pattern of the DEMs.
FABDEM and its underlying source data, the Copernicus DEM, show similar large scale patterns of areas at risk of future
SLR. On a local scale, in central parts of the lower delta, where vegetation accompanies the channel network (mainly trees),
the Copernicus DEM tends to underestimate the potentially drowned area and would also underestimate the impact of a flood

with a critical surge height (Fig. S26). In contrast, if the FABDEM is used to estimate flood impact, calculations must be taken
cautiously when focusing on urban settlements in the delta as building infrastructure constitutes certain resistance against
flooding, which is neglected by the DTM (see also Bellos and Tsakiris, 2015). Discrepancies between DSM (Copernicus DEM)
and DTM (FABDEM) are most prominent in Yangon City. Here, the delineated area below sea level must not be equalised
with the area affected by flooding due to SLR and other mechanisms since built-up infrastructure has been neglected. Indicating

the lowest elevations, the CoastalDEM v2.1 shows the most extensive area below future sea level. Though considering building
infrastructure indirectly via population data, on a local scale, CoastalDEM v2.1 reveals a significantly different pattern for
Yangon City than FABDEM (Fig. S26). We interpret this discrepancy as reflecting that population data as an indicator for
urban development and thus positive error in DEMs (Kulp and Strauss, 2018) does not provide a thoroughly proper approach
to improve the correction of vertical error in the underlying NASADEM. However, to translate this discrepancy between

CoastalDEM v2.1 and FABDEM into population at risk in these areas would require further and detailed investigations on
township and ward levels which would be – in view of the 74 townships in the delta – beyond the scope of this study.

Given the absence of any other independent ground truthing possibilities and no DEM clearly outperforming the others
throughout all validations and statistics, we consider the FABDEM and CoastalDEM v2.1 as the most consistent ones with





respect to elevation indicated by spot heights and the AD-DEM. Though amongst these, CoastalDEM v2.1 reveals in some

places slightly better absolute statistics, they are often in the same range for the FABDEM and the latter, in addition, outperforms the CoastalDEM v2.1 in terms of spatial resolution and the absence of sensing artefacts. Thus, we suggest the FABDEM as the most suitable when it comes to consider the role of topography for flood risk assessment in the Ayeyarwady Delta. To further improve the FABDEM for the Ayeyarwady Delta, we suggest supplementing it with the local elevation data used for the AD-DEM, which will complement the large scale suitability of the AD-DEM by allowing small scale studies at

high vertical accuracy.

**5.2 Implications for assessing flood risk in the Ayeyarwady Delta**

By integrating 10 global DEMs in addition to the local AD-DEM to assess the impact of SLR on the Ayeyarwady Delta, we visualise and quantify the range of potential consequences in terms of area and population falling below future sea level. In addition to the uncertainties, we quantified related to the topographic data, further uncertainties exist with regard to the

population and SLR datasets used and that process dynamics in reality are more complex than could be captured by our study. Given that our calculations are based on population estimates of 2020, these do not consider any trends such as urbanisation and population growth which has occurred (Vogel et al., 2022) and can also be expected to continue in the future. Thus, the number of people below sea level may be considerably larger in the future than estimated in this study. Furthermore, subsidence will increase the impact of SLR, lowering elevation locally by up to ~2.70 m until 2150 if continuing at rates of 10–20 mm yr$^{-}$

$^{1}$ as observed for Yangon City (van der Horst et al., 2018). However, subsidence rates, especially in deltaic settings, are caused by many different processes that can be highly spatio-temporal variable (Shirzaei et al., 2021) and increased human activities are known to accelerate rates (Minderhoud et al., 2017; 2018; Candela and Koster, 2022), hampering reliable long term estimations. Therefore, we highlight that the IPCC SLR scenarios – although projected for 2100 and 2150, respectively – could be realised much sooner, and land elevation thus becomes even more critical. The lack of knowledge on the magnitude of

subsidence processes in the Ayeyarwady Delta, together with the limited availability of information and high quality local data related to subsidence drivers like groundwater withdrawal (Viossanges et al., 2017), adds unknown and potentially large uncertainties to our SLR impact assessments. For example, groundwater withdrawal has been monitored only sporadically at artesian irrigation projects and in urban areas (Viossanges et al., 2017). However, some observations indicate falling groundwater levels and according to the Water Supply System Master Plan (JICA, 2014), up to 3 million people rely on

groundwater resources in Yangon City. Recent studies of Hashimoto et al. (2022) document even larger extraction amounts than reported by JICA (2014), which are also very high compared to other Asian cities affected by extraction induced subsidence (Hashimoto et al., 2022). Though agricultural areas in the lower delta are reported to rely less on groundwater than townships of the upper delta and Yangon City, reliable estimates are absent, while groundwater use for cultivation (e.g., horticulture, aquaculture, etc.) is unknown (Viossanges et al., 2017). Thus, rates of RSLR will most likely be even larger than

estimated so far. On a global scale, Tay et al. (2022) came to similar conclusions for several coastal cities, including Yangon City, that the impact of land subsidence is underestimated by recent RSLR assessments like the IPCC 6[th] Assessment Report.



Consequently, future RSLR may affect substantially more land and inhabitants than we could estimate for the Ayeyarwady Delta so far, especially in the eastern part of the delta. Similarly, the RSLR rates of Nicholls et al. (2021) provide an approximation of future RSLR along the delta coast that is still associated with a range of uncertainties due to the lack of subsidence information covering the entire delta and the use of simplified, linearly extrapolated subsidence rates.

To gain a more complete understanding of flood risk in the delta, it is important to consider the different mechanisms and hydrodynamics that are involved in or add to SLR. Beside land subsidence lowering the elevation of the land and hereby increasing RSLR (Minderhoud et al., 2020; Nicholls et al., 2021), these include interactions with tidal constituents, surge, and waves as well as potential feedbacks on nearshore bathymetry and shoreline morphology (Idier et al., 2019). Together with local mechanisms such as drainage backflow and groundwater inundation that have been observed especially in urban areas (Habel et al., 2020), they will make the flood pattern more complex. In tropical deltas characterised by a monsoonal climate, such as the Ayeyarwady, pluvial flooding and increased river discharge constitute sources of flood risk nearly independent from the coast but rather coming from the hinterland (Figs. 5 and S13–S18), affecting also higher elevations (Fig. S17) and thereby enlarging inundation areas impacted by sea-borne flood hazards. Data paucity challenges the simulation of these so called multi-mechanism or compound floodings in the Ayeyarwady Delta and would require the acquisition and careful selection of further (available) data. As input elevation data for simulating (compound) flooding at delta scale, we found the FABDEM to be the best currently available DEM, in the absence of superior high resolution DEMs. The FABDEM may be further improved by integrating local elevation data from this study. To simulate floods adequately in urban settings such as Yangon City, the FABDEM is recommended to be used either with reinserted building heights from the Copernicus DEM or in combination with increased surface roughness (i.e., via increasing the Manning's coefficient or via the friction term within the momentum equations; Schubert and Sanders, 2012; Bellos and Tsakiris, 2015). In the absence of these flood models, we recommend to revise existing flood hazard assessments by integrating validated data (such as accurate DEMs) and mappings of past flood events in order to consider multiple flood risk when designing present and future sustainable development pathways for the Ayeyarwady Delta. In this sense, we also call for the acquisition and accessibility of accurate and high resolution data, such as LiDAR data, to the scientific public to reduce uncertainties of flood risk assessments on smaller scales, thereby supporting local risk mitigation and adaptation strategies. Based on the assessment of land elevation in view of SLR and delta flooding in the Ayeyarwady Delta, low lying and densely populated areas within the most seaward districts should be addressed in particular by developing risk mitigation and adaptation strategies, while also more inland delta population should be prepared to face a higher risk of flooding due to sea level rise in the next ~100 years.

## 6 Conclusions

As flood risk assessments based on high accuracy elevation data either do not exist or are not publicly accessible for the Ayeyarwady Delta in Myanmar, available risk assessments so far rely at best on global DEMs, suffering from low vertical accuracy and remote sensing artefacts. To investigate their uncertainties and thus their suitability to provide reliable





information on flood risk due to the impact of SLR and other water related hazards, we assessed the land elevation of the

Ayeyarwady Delta in relation to continuous local MSL by (a) generating the new, local AD-DEM based on topographical map elevation data, and (b) comparing the performance of the new AD-DEM and 10 available global DEMs, referenced to local MSL. Thirdly, (c), we identified deltaic areas prone to SLR and monsoon flooding and interpreted their relation to topography. With respect to our research questions, we conclude that

(i) The new, local AD-DEM, generated based on digitised spot heights and supplemented by elevation points extracted

from contour lines in particularly data sparse regions, indicates a mean delta plain elevation of ~2.1 m (median ~1.3 m) above local MSL and is accurate for elevations ≤ 10.0 m at 1.0 m RMSE. With a horizontal resolution of 750 m, it is suitable for studying topography on delta scale and provides an independent dataset to validate commonly used global elevation data.

(ii) The global DEMs integrated into this comparison show up to several metres deviation from geodetic spot heights and

the AD-DEM. ASTGTM v003 reveals the lowest vertical accuracy and significantly overestimates low-lying areas (RMSE of up to 9.20 m), while ACE2 performs comparably well in comparison with spot heights but overall indicates significantly lower elevations than AD-DEM and all other global DEMs (mean height residuals of -0.47 to -1.87 m). FABDEM and CoastalDEM v2.1 show the highest correspondence with AD-DEM and low elevation spot heights with height residuals at centimetre to decimetre scale and RMSE of 1.75 to 2.00 m.

(iii) Presently available flood risk assessments of Myanmar often either do not refer to underlying elevation data or use global SRTM and ALOS DEMs, which, however, suffer from sensing artefacts, overestimate delta elevation compared to independent elevation data and thus underestimate areas at risk of SLR. As DEMs corrected or trained with additional datasets perform substantially better, they allow for delineating areas at risk of SLR in higher resolution than AD-DEM. Using these DEMs (i.e., AD-DEM, FABDEM, and CoastalDEM v2.1) and assuming high

GHG emission scenarios (where vertical land motion is likely underestimated), 30.7 to 46.7 % of the total delta area with at present 1.7 to 2.9 million people (2020 population) will fall below sea level by 2100, most of them in Pyapon, Labutta, and Myaungmya districts. By 2150 (probably sooner), up to 67.5 % of the delta could be drowned, an area with at present 5 million people that also includes more inland districts. As monsoon flooding affects also higher elevations, it has the potential to further enlarge inundation areas impacted by sea-borne flood hazards.

(iv) To further investigate the impact of SLR and delta flooding in the Ayeyarwady Delta in the absence of high resolution DEMs, we recommend the use of the FABDEM as it proves to perform comparably well, is of adequate spatial resolution and does not suffer from stripes or other artefacts. To simulate floods also adequately in urban settings such as Yangon City, the FABDEM is recommended to be used either with reinserted building heights from the Copernicus DEM or in combination with increased surface roughness. Further improvement may include

supplementing the FABDEM with local elevation data from this study. For DEM accuracy assessments in general, we recommend to analyse a given dataset by as many quality standards as possible to gain a holistic picture, and to





determine the most accurate one based on its overall performance instead of relying on individual, selected accuracy parameters.

(v)   Our findings indicate that water related hazards in the Ayeyarwady Delta and available risk assessments need to be re-examined by integrating validated elevation data to strengthen their reliability. As SLR may be compounded by other types of flooding and involves numerous mechanisms and process dynamics, which are themselves subject to uncertainties, the overall flood pattern becomes more complex. With our study, we addressed the uncertainties related to DEMs and land elevation (apart from vertical land motion), and stress the need of addressing low lying and densely populated areas within the most seaward districts in particular by developing risk mitigation and adaptation strategies, while also more inland delta population should be made aware to face a higher risk of flooding due to SLR in the next ~100 years.

**Data availability**

The AD-DEM will be made available in the Pangaea online data repository. Digitised elevation data from topographical maps will be made available upon request.

**Author contributions**

K.S., P.S.J.M. and D.B. conceptualised the study. K.S. acquired the elevation data and topographic maps. K.S. and A.P. conducted digitisation. K.S. performed processing, data analyses and visualisation. K.S. and P.M. discussed intermediate and final results. D.B., H.B. and F.K. acquired funding and are responsible for supervision and project administration. K.S. prepared the original draft of the manuscript. All co-authors reviewed and edited the manuscript.

**Competing interests**

The authors declare that they have no conflict of interest.

**Acknowledgements**

We thank SIS Maps of the Map Department of the *Staatsbibliothek zu Berlin* for providing topographical map sheets including related information. Regine Spohner from University of Cologne is thanked for georeferencing the topographic map sheets. TanDEM-X data and the Copernicus DEM were provided by DLR upon a science proposal (DEM_HYDR3351). Sinem Ince and Saskia Esselborn from GFZ are gratefully acknowledged for their kind advice on the geoid offset calculation. We thank the authors of the IPCC AR6 Sea Level Projection Tool for developing and making the SLR projections available, multiple funding agencies for supporting the development of these projections, and the NASA Sea Level Change Team for developing





and hosting the tool. This research was financially supported by a grant from the German Research Foundation (DFG project
number 411257639; BR 5023/4-1; BR 877/37-1; KR 1764/28-1).

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
