# Peer review of "Assessing land elevation in the Ayeyarwady Delta (Myanmar) and its relevance for studying sea level rise and delta flooding"

_EGUsphere, 2022_

## Author Response (AR1)

**Author's response to comments on "Assessing land elevation in the Ayeyarwady Delta (Myanmar) and its relevance for studying sea level rise and delta flooding" by Katharina Seeger et al.**

*Dear Editor, Dear Reviewers,*

*Thank you very much for your detailed reviews of our submission, your positive feedback and helpful comments as well as for handling our paper through the interactive review process. We respond to the required adjustments point by point and mark our responses in italics after each reviewer comment. All revised files are uploaded, including a track-change version.*

*Thank you again for your valuable feedback, which improved the manuscript.*

*Best regards,*

*Katharina Seeger (on behalf of the authors)*

**Response to referee comment #1**

abstrac, line 25: I wouldn't use "open source" here, but just "open", as open source feels more appropriate for software code (the source code that is open).

*Changed as requested.*

usage of colors in figures. I feel that the figures have quite a lot of information, and showing plots with several categories represented only by colors, is always a challenge. In figure 5, for instance, is very difficul to identify the lines for each DEM. In the graphical abstract, we have 10 (?) lines all with the same symbols, and I have doubts if people with color-impaired vision will be able to see all the lines. I suggest the authors to re-think these stilistic choices.

*Agreed. The symbology of the figures with colourful plots was adjusted by using a combination of colour and symbol type.*

In fig.5, also try to rename the sub-figures so they have the same letter as the profile, because "fig.5D, profile C", etc is a bit confusing. In my opinion, the enlaged part of profile A does not help much...

*We agree and changed the labeling of subfigures and adjusted the enlarged part of profile A to focus on the lowest elevations, i.e., 5 m above MSL.*

In figure 1, I would rather see a larger version of the DEM (maybe with a shaded relief as texture), as large as the page, because that is very important. The insets can be repositioned, the colorscale could be continuous (and use less space), and the map of points could be a small map of point density, since that its the message (and you have a version of that map in the supplemental material). The larger DEM should also bring the names of places mentioned in the text, as we readers are usually not so familiarized with the area.

*We adjusted Fig. 1 by enlarging subfigure (a) and adding features like latitudes and longitudes to support our statements of ll. 300–306 (i.e., ll. 312–320 in the revised manuscript). We deliberately did not provide further information on deltaic settlements in Fig. 1a as otherwise, the most relevant information, i.e., the presentation of the new AD-DEM would not become clear. However, we considered to add key references and improved in-text referencing to Fig. S25 (i.e., Fig. S26 in the revised manuscript) which contains the location of all settlements (e.g., in ll. 511 ff. (ll.533 in the revised manuscript)). Subfigures (b) to (d) have been put in the supplementary material.*

Another choice that I questioned as I was reading was to show ACE2 results instead of, say, ALOS AW3D30. It seems to me that AW3D30 has a much larger user community than ACE2.

*We agree and to serve the international userbase of AW3D, we added the AW3D30 to the other DEMs in the main text and restructured the figures accordingly. We also provide results for the ACE2 in the main text to show the entire range of DEM performance, where satellite-based DEMs not only overestimate but also underestimate local elevation data.*

In figure 3, we can see some major differences between the DEMs, but not the artifacts mentioned in the text. A slope map would be much better for this.

*We created a slope map as suggested but the artefacts are hardly visible, even after applying different symbologies. Please see below the slope map for the AW3D30, which suffers from stripes.*

[Figure]

abouth the methods:

the authors could be clearer on why they digitized point heights for the lower areas and contours for the higher areas. From fig.1, I can see that, being a very flat area, there are not much more topographical information in the maps besides spot height for these low-lying areas, but that is not very clear in the text.

*We rephrased the respective methodological section and provided a more detailed explanation on the presence and use of spot height and contour data. Please see also our reply below.*

in line 180 you say: "From these, points were extracted at 250 m and 2000 m intervals, respectively" what do you mean? I don't understand exactly what these two values represent. Does it mean that the distance between vertices of the digitized lines are 250m apart? why? what is the reationale behind this? I see that you created DEMs from these two datasets, but I still don't see the reason for them (I would have done only one version)

*We agree with the reviewer that the sentence may be confusing. We tested whether the selection of the interval at which points were extracted from the contour lines had an impact on the DEM interpolation. We adjusted the section for clarification: "Point elevations are regularly distributed in the central to southern delta parts, while they are more unsystematically and less densely distributed especially in the northern delta parts. Elevation contours are only present in the upper delta, i.e., north of 17.0° N. As contour lines already constitute indirect elevation information instead of locally precise ground measurements, we considered them only in areas that are not spatially covered by spot height data in order to supplement the input elevation data for DEM interpolations. From these contour data, points were extracted at 250 m and 2000 m intervals, respectively, to test whether the spacing has an impact on DEM interpolation."*

line 189 - what was the resolution? say it here and don't leave your reader wondering.

*Agreed. We added the spatial resolution: "The grid cell resolution of 750 m × 750 m was defined as justified by the point density following Aguilar et al. (2006)."*

lines 300-306 - the details mentioned in the text can't be easily seen on the map of fig.1. you need a larger map for that

*We addressed this issue by enlarging Fig. 1 and adding latitudes and longitudes.*

line 317 you say "whereas topography as represented by AW3D30 is affected by distortions that reflect the sensor swaths" - what sensor swaths? SRTM has those artifacts caused mostly by the mast oscillation, but aw3d30 is built by photogrammetry of also images.

*We agree with the reviewer that the wording on the artefacts of AW3D30 is improper. We corrected and rephrased the sentence: "Both SRTM and AW3D30 suffer from remote sensing artefacts as SW–NE oriented stripes are visible in the SRTM, whereas topography as represented by AW3D30 is affected by distortions that reflect the swaths of the ALOS PRISM (triplet mode; Fig. 3)."*

**Response to referee comment #2**

The authors present very relevant work by assessing the quality of elevation maps for a data scarce country like Myanmar and making improved datasets available. It would be helpful if the authors could add the link to Pangea data repository as well as a data when the data will become availble.

*We fully agree and both the AD-DEM and the FABDEM, which was adjusted to the local spot height delta in the central and southern parts of the Ayeyarwady Delta, have been submitted to an online data repository and will be published as soon as the paper is published. DOIs and references to these datasets were added in the main text and in the list of references.*

In general the research seems sound yet at some points difficult to read. I feel that the work would benefit from a more comprehensive overview of the source data of the different elevation models. This would also allow a more systematic discussion of the artefacts.

*We added several sentences in the main text to address the most important information on acquisition technique (including eventual edits) and DEM type and added the respective references in the list of references:*

*"Both SRTM and TanDEM-X are based on radar interferometry while ASTGTM and AW3D30 were processed from optical imagery (Farr et al., 2007; Tachikawa et al., 2011; Tadono et al., 2016; Wessel, 2018). All these DEMs are digital surface models (DSMs). By including altimetry data, ACE2 (nearly a digital terrain model (DTM)) was generated out of SRTM whereas CoastalDEM v2.1 (nearly DTM) is a correction of the SRTM-based NASADEM, which was trained with ICESat 2 satellite LiDAR data and further datasets in a neural network (Berry et al., 2010; Kulp and Strauss, 2021). Recently, edits on the TanDEM-X data have led to the release of the Copernicus DEM (DSM; Airbus Defence and Space, 2020). By eliminating the bias of tree and buildings heights, Hawker et al. (2022) generated the FABDEM (nearly DTM) out of the Copernicus DEM. In contrast, the GLL-DTM v1 constitutes a DTM that was interpolated from ICESat 2 satellite LiDAR data (Vernimmen et al., 2020)."*

*In addition, in ll. 201–203 (i.e., ll. 214–215 in the revised manuscript) we refer to a comprehensive overview of the DEMs and their source data included in the supplementary material. This information is not included in the main text to not further extend the length of the manuscript (see also comment of reviewer 1).*

The comparison with flood data is interesting yet it is difficult to get a quantitative understanding of the correlation between floods and depressions from Figure 5.

*We agree with the reviewer that Fig. 5 does not provide quantitative information about the relation between elevation and flooding. However, in view of the profile lengths and the size of*

*individual inundated areas, further differentiation between the flood events would often not be visible in the figure as they would overlap. In contrast, shorter profile sections would not capture the performance of the DEMs in the different delta parts adequately. As our study focusses on the entire Ayeyarwady Delta, we have chosen the first option. Additionally, we addressed the relation between elevation and the respective flood events in figure S17 (i.e., Fig. S18 in the revised manuscript).*

Few other comments/suggestions:

- Line 76-79: reference needed
  *We added a reference by referring to Üstün et al. (2016). The reference has been added in the list of references as well.*

- Figure 1c and d: improve readability
  *As the original quality of the topographic maps does not allow for better readability, we added enlargements of these subfigures into the supplementary material.*

- Line 213: add date of retrieval
  *Added as requested.*

- Line 244: repeat reference to sampling technique
  *We added a reference as requested by referring to ESRI (2023). The reference has been added in the list of references as well.*

- Line 250: Why is AW3D excluded?
  *In order to serve the international userbase of AW3D, we added the AW3D30 to the other DEMs in the main text and restructured the figures accordingly.*

- Line 261: "on investigations" can you explain this more specific?
  *Changed as requested. As we investigated the precipitation and discharge data for their maximum peaks, we replaced "investigations" by "maximum": "The pre-event imagery was recorded in February during the dry season while the selection of imagery capturing maximum flood extents was made based on maximum rainfall amounts and river discharge."*

- Line 264: which satellite microwaves are used?
  *As this question addresses the source data of a data product we used, we think that further information regarding the details of this dataset can be looked up in the respective documentation to which we already referred (i.e., Brakenridge et al., 2022).*

- Line 302: what imperfection?
  *The imperfection lies in the sharp, unrealistic elevation change we describe in the sentence before. However, we specified this imperfection.*

- Line 325: How where trees and buildings removed?

  *As this question addresses the processing of a data product we used, i.e., the FABDEM, we think that further information regarding the processing details of this dataset can be looked up in the respective publication. However, to clarify that the correction for tree and building heights was done by the developers of the FABDEM, we added an extra reference to their publication.*

- Line 341 and further: are the 2 numbers behind the comma a meaningful level of significance?

  *We decided to provide two decimals to highlight the – sometimes slight – differences between the DEMs. In addition, the second decimal is relevant in view of $R^2$. Finally, several other studies that investigate the accuracy of DEMs provide their statistics on centimetre scale (e.g., Erasmi et al., 2014 (https://doi.org/10.3390/rs6109475), Rao et al., 2014 (https://doi.org/10.5194/isprsannals-II-8-187-2014); Hawker et al., 2019 (https://doi.org/10.1016/j.rse.2019.111319); Pasquetti et al., 2019 (https://doi.org/10.3390/rs11151767); Kulp and Strauss, 2021 (https://assets.ctfassets.net/cxgxgstp8r5d/3f1LzJSnp7ZjFD4loDYnrA/71eaba2b8f8d642d d9a7e6581dce0c66/CoastalDEM_2.1_Scientific_Report_.pdf); Hawker et al., 2022 (https://doi.org/10.1088/1748-9326/ac4d4f)).*

- line 349: is a R2 of 0.85 also good given the spatial correlations/dependencies in the data?

  *Of course, it is a point of discussion whether to assume an $R^2$ of 0.86 as good or not. As we only provide the statistical results in this section, our intention with this sentence was to document that the $R^2$ of ACE2 is in that good as it constitutes the maximum of all datasets that were investigated. However, this does not preclude that other error metrics like MAE, RMSE, etc. indicate a worse performance of ACE2 compared to the other DEMs. To avoid misunderstandings when using the term 'good', we rephrased l. 349 (i.e., ll. 364–365 in the revised manuscript): "In contrast, while the corrected SRTM version ACE2 has a comparatively high correlation ($R^2$ = 0.86), it mainly shows negative deviations (Table 1; Fig. S7). The absolute error is still more than 3 m, and the elevation data shows substantial scatter as reflected by the maximum standard deviation σ compared to all DEMs."*

- Line 424: what is meant with this sentence?

  *We agree that the wording of this sentence is not very explicit rephrased it accordingly: "The 2015 and 2020 monsoon floodings show different spatial extents, with the 2020 flooding affecting more areas in the southern part of the delta than in 2015. These areas are relatively lower in elevation as documented by the DEMs (Fig. S18; further details are given in the supplementary material)."*

- Line 644: coastal squeeze may be mentioned explicitly

*We added coastal squeeze and added the respective references in the list of references: "Beside land subsidence lowering the elevation of the land and hereby increasing RSLR (Minderhoud et al., 2020; Nicholls et al., 2021), these include interactions with tidal constituents, surge, and waves, potential feedbacks on nearshore bathymetry and shoreline morphology (Idier et al., 2019) as well as consideration of coastal squeeze (Luo et al., 2018). Together with local mechanisms such as drainage backflow and groundwater inundation that have been observed especially in urban areas (Habel et al., 2020), those processes will make the flood pattern more complex."*